# Neutralization of p40 Homodimer and p40 Monomer Leads to Tumor Regression in Patient-Derived Xenograft Mice with Pancreatic Cancer

**DOI:** 10.3390/cancers15245796

**Published:** 2023-12-11

**Authors:** Monica Sheinin, Susanta Mondal, Kalipada Pahan

**Affiliations:** 1Department of Neurological Sciences, Rush University Medical Center, Chicago, IL 60612, USAsusanta_mondal@rush.edu (S.M.); 2Division of Research and Development, Jesse Brown Veterans Affairs Medical Center, Chicago, IL 60612, USA

**Keywords:** pancreatic cancer, pancreatic ductal adenocarcinoma, IL-12p40 monomer, IL-12p40 homodimer, patient-derived xenograft mouse model, immunotherapy, IL-23, IL-12, apoptosis, cell death

## Abstract

**Simple Summary:**

Pancreatic ductal adenocarcinoma (PDAC) is one of the deadliest cancers, and immunotherapy could be one of the safest options to control PDAC. Although the p40 family of cytokines has four members comprising IL-12, p40 homodimer (p40_2_), p40 monomer (p40), and IL-23, we found an increase in p40_2_ and p40 and a decrease in IL-12 and IL-23 in serum of pancreatic cancer patients as compared to healthy controls. Similarly, human pancreatic cancer cells also produced greater levels of p40_2_ and p40 and lower levels of IL-12 and IL-23 than normal human pancreatic cells. Accordingly, neutralization of p40_2_ by a3-1d monoclonal antibody (mAb) and p40 by a3-3a mAb separately led to apoptosis of pancreatic cancer cells and tumor shrinkage in a patient-derived xenograft (PDX) mouse model of pancreatic cancer, highlighting the possible therapeutic potential of mAbs against p40_2_ and p40 in pancreatic cancer.

**Abstract:**

Pancreatic cancer is a highly aggressive cancer with a high mortality rate and limited treatment options. It is the fourth leading cause of cancer in the US, and mortality is rising rapidly, with a 12% relative 5-year survival rate. Early diagnosis remains a challenge due to vague symptoms, lack of specific biomarkers, and rapid tumor progression. Interleukin-12 (IL-12) is a central cytokine that regulates innate (natural killer cells) and adaptive (cytokine T-lymphocytes) immunity in cancer. We demonstrated that serum levels of IL-12p40 homodimer (p40_2_) and p40 monomer (p40) were elevated and that of IL-12 and IL-23 were lowered in pancreatic cancer patients compared to healthy controls. Comparably, human PDAC cells produced greater levels of p40_2_ and p40 and lower levels of IL-12 and IL-23 compared to normal pancreatic cells. Notably, neutralization of p40_2_ by mAb a3-1d and p40 by mAb a3-3a induced the death of human PDAC cells, but not normal human pancreatic cells. Furthermore, we demonstrated that treatment of PDX mice with p40_2_ mAb and p40 mAb resulted in apoptosis and tumor shrinkage. This study illustrates a new role of p40_2_ and p40 monomer in pancreatic cancer, highlighting possible approaches against this deadly form of cancer with p40_2_ and p40 monomer immunotherapies.

## 1. Introduction

Pancreatic cancer is a highly malignant cancer commonly detected at advanced stages. It accounts for about 3% of cancers in the US, with the lowest 5-year relative survival rate at 12% [1]. Pancreatic ductal adenocarcinoma (PDAC) is the predominant neoplasm responsible for the majority of pancreatic cancer cases, and many patients have unresectable, advanced, or metastatic manifestation at diagnosis [2]. Current treatments including surgery, radiation, targeted therapy, and chemotherapy have shown inadequate improvement in survival [3]. Recently, studies have shown that immunotherapy is a worthwhile approach, which can enhance the immune system’s capacity to identify and kill cancer cells [4,5]. Some limitations of existing therapies include chemoresistance, a dense desmoplastic tumor microenvironment, genetic heterogeneity across patients, and a lack of specific biomarkers [6,7].

The p40 family of cytokines has four members, p40 homodimer (p40_2_), p40 monomer (p40), IL-12 (p40:p35), and IL-23 (p40:p19) [8]. IL-12 plays a crucial role in the early inflammatory response by producing T helper type 1 (Th1) cells [9,10]. It is believed that most effects are mediated by IFN-γ secretion [11]. Until recently, p40_2_ and p40 were considered inactive partners of the IL-12 family [12,13,14]. Since all four members of the p40 family contain p40 in distinctive forms, a specific monoclonal antibody (mAb) is necessary to characterize these molecules. Monoclonal antibodies (mAbs) are immunoglobulins produced to target specific proteins and found to improve long-term outcomes of patients [15]. Antibody-mediated immunotherapy targets tumors using mAbs to inhibit oncogenic signaling, immune suppression, or immune checkpoint blockade [16]. Therefore, we have produced neutralizing mAbs against p40 and p40_2_ [17] and, by using such mAbs, we have delineated that the level of p40 is higher in the serum of prostate [18] and breast cancer [19] patients as compared to healthy controls, and selective neutralization of p40 by mAb leads to tumor regression in mice [18,19]. On the other hand, the level of p40_2_ is reduced in the serum of prostate and breast cancer patients in comparison with healthy controls [18,19], indicating the specificity of our finding.

In contrast to prior results, we found an increase in both p40_2_ and p40 in the serum of pancreatic cancer patients when matched to healthy controls and in various human pancreatic cancer cells in comparison to normal human pancreatic duct epithelial cells. Furthermore, we demonstrated that selective neutralization of p40_2_ by p40_2_ mAb a3-1d and p40 by p40 mAb a3-3a stimulated death in human pancreatic cancer cells in culture and in vivo in PDX tumor tissue. This study highlights the role of p40_2_ and p40 in the survival of pancreatic cancer cells and tissues that could be of therapeutic benefit for patients with pancreatic cancer.

## 2. Materials and Methods

### 2.1. Cell Culture and Reagents

Human pancreatic cancer cell lines were purchased from ATCC (Manassas, VA, USA). PANC-1 (CRL-1469) were cultured with DMEM (10-017-CV). BxPC-3 (CRL-1687) were cultured with RPMI-1640 (11875-093) (ThermoFisher Scientific, Waltham, MA, USA). Both cell lines were supplemented with 10% FBS obtained from Atlas Biologicals (Fort Collins, CO, USA) and antibiotic–antimycotic from ThermoFisher Scientific, Waltham, MA, USA). Human pancreatic duct epithelial cells HPDE-H6c7 (ECA001-FP) purchased from Kerafast (Boston, MA, USA) were cultured with Keratinocyte SFM (Invitrogen, Waltham, MA, USA) supplemented with EGF and bovine pituitary extract as well as antibiotic–antimycotic. All cell lines were cultured at 37 °C and 5% CO_2_ in complete medium. When cells grew to 80% confluency, they were passaged. Phosphate-buffered solution (PBS) was added to the flasks to wash cells, then they were treated with 0.25% trypsin from Corning. Culture medium neutralized the trypsin, cells were resuspended then seeded into T75 flasks, and the cells from the logarithmic phase were utilized for experiments. Additional cell line details are included below (see Table 1).

### 2.2. Serum Samples of Pancreatic Cancer Patients

The serum samples of pancreatic cancer patients before drug treatment and some age-matched healthy controls were purchased from Discovery Life Sciences, Los Osos, CA, USA. Serum samples of the rest of the age-coordinated healthy individuals were obtained from Rush University Medical Center. Controls are defined as healthy individuals without significant medical history. Details are included below (see Table 2).

Serum samples were purchased mainly from Discovery Life Sciences (DLS). Some normal samples were obtained from Rush University Medical Center (RUMC). Each sample was analyzed for p40, p40_2_, and IL-12 three times by ELISA.

### 2.3. Sandwich ELISA

Sandwich ELISA was performed to quantify p40_2_ and p40 as described by our laboratory [17,18,20]. Please see dilutions for p40_2_ mAb and p40 mAb capture and biotinylated detection antibodies [19]. Concentrations of IL-12 and IL-23 were measured in patient serum or tissue homogenates via ELISA (eBioscience/Thermo Fisher (San Diego, CA, USA)), according to the manufacturer’s guidelines as described previously [18].

### 2.4. Assessment of Cell Viability: MTT and LDH Assays

Cell viability was calculated using the 3-(4,5-dimethylthiazol-2-yl)-2,5-diphenyl tetrazolium bromide (MTT) method (Sigma) as delineated in previous experiments [18,19]. Cells were seeded in 12-well plates with 500 µL of DMEM and RPMI-1640 medium for 24 h then treated in serum-free medium. After cells were treated for 24 h, 100 µL supernatant was separated to be used for the LDH assay followed by addition of MTT reagent. To measure mitochondrial activity, formazan crystals were dissolved by adding equivalent volume of solubilization solution. After treatment, 300 μL of culture medium was removed from each well and 10 μL of MTT solution (5 mg/mL) was added and incubated for 30 min. Supernatant was added to a 96-well plate, then absorbance was measured at 595 nm with the Thermo Fisher Multiskan MCC plate reader. Lactate dehydrogenase (LDH) activity was measured using an LDH assay kit (Sigma, St. Louis, MO, USA) as outlined before [18,19].

### 2.5. Fragment End Labeling DNA—TUNEL Assay

Fragmented DNA was detected in cells and tissues by the terminal deoxynucleotide transferase (TdT)-facilitated binding to 3′OH ends of DNA fragments broken due to apoptotic signals, using a kit (TdT FragEL DNA Detection Kit) from Millipore Sigma (Burlington, MA, USA) as described previously [18,19]. PANC-1, BxPc-3 and HPDE cells were cultured to 70–80% confluence, and plated on coverslips, then fixed with chilled methanol (Fisher Scientific, Waltham, MA, USA) for an hour, followed by rinses with sterile PBS. Cells and tissues were treated as described by us in previous studies [21]. Coverslips were mounted, and dried overnight. Then, slides were imaged under the Olympus BX41 with an attached Hamamatsu ORCA-03G camera.

### 2.6. Flow Cytometry

For FACS analysis, single cells were resuspended and isolated using a cell strainer from PANC-1, BxPC-3, and HPDE cells. They were stained using an apoptosis kit for flow cytometry (Thermo Fisher, Waltham, MA, USA) as described previously [22]. Briefly, single-cell suspensions of PANC-1 and BxPC-3 cells were stained with Zombie Aqua™ Fixable Viability Kit (Biolegend, San Diego, CA, USA). Cells were washed with FACS buffer (Thermo Fisher), stained with PE-anti-human IFNγ antibody (Biolegend), and identified by FACS Canto II Flow cytometer (BD Biosciences) and analyzed using FlowJo Software (v10). Controls were composed of Annexin V, PI, and unstained cells.

### 2.7. Immunostaining

Coverslips containing PANC-1, BxPc-3, and HPDE cells were cultured to 70–80% confluence. The cells were fixed with chilled methanol (Fisher Scientific, Waltham, MA, USA) for one hour, then washed with filtered PBS. For tissue samples, antigen retrieval in heated sodium citrate buffer pH 6.0 multiple times was necessary [23]. A detailed procedure for immunostaining was previously outlined by us [21]. The primary antibodies used included IFN-γ (1:100; Thermo Fisher, Waltham, MA, USA), cleaved caspase 3 (1:400; Cell Signaling, Danvers, MA, USA), CA19-9 (1:500; Thermo Fisher, Waltham, MA, USA); they were incubated for 2 h on a shaker at RT (see Table 3). Cells and tissues were rinsed multiple times in filtered PBS, then incubated with Cy5-labeled secondary antibody (1:200; Jackson ImmunoResearch, West Grove, PA, USA) for 1 h. After three washes in PBS, cells were incubated for 5 min in 4′, 6′- diamindino-2-phenylindole (DAPI, 1:10,000; Millipore Sigma, Burlington, MA, USA) followed by mounting, drying overnight, and observing under the Olympus BX41 equipped with a Hamamatsu ORCA-03G camera.

### 2.8. Real-Time PCR

Real-time PCR was performed as described [20,22] in the ABI-Prism 7700 sequence detection system (Applied Biosystems, Foster City, CA, USA) using PowerUp SYBR green master mix (Thermo Fisher, Waltham, MA, USA). The mRNA expression of the targeted gene was standardized to the level of GAPDH mRNA. The following primers were used for human IFN-γ and GAPDH.

IFN-γ:

Sense: CGGCACAGTCATTGAAAGCC

Antisense: TGCATCCTTTTTCGCCTTGC

GAPDH:

Sense: GCATCTTCTTGTGCAGTGCC

Antisense: TACGGCCAAATCCGTTCACA

#### Immunoblotting

Immunoblotting was conducted as described by us [20,22]. The tissues were lysed and protein extracted using a detailed method described by us previously [21]. Protein concentration was analyzed from the lysed tissue supernatant, using the Bradford method (Bio-Rad, Hercules, CA, USA). Additional immunoblotting procedure was outlined by us in previous studies [21]. The membrane was blocked for 1 h in intercept blocking buffer (Li-COR, Lincoln, NE). After, the membranes were incubated overnight at 4 °C on a shaker in primary antibodies diluted in blocking buffer at indicated dilutions (see Table 3). Actin was run as a loading control. The following day, membranes were washed in PBS containing 0.1% Tween 20 (PBS-T) for 30 min (3× for 10 min), and incubated with secondary antibodies (Li-COR, Lincoln, NE; Dilution: 1:10,000) for 1 h at room temperature. Then, the blot was washed in PBST for 30 min and imaged using the Odyssey Infrared Imaging System (Li-COR, Lincoln, NE, USA).

### 2.9. Experimental Animals

Female NOD scid gamma (NSG) mice engrafted with patient-derived pancreatic cancer were generated by The Jackson Laboratory (#TM01212) [16,24,25]. Mice were engrafted subcutaneously in the right flank. All NSG mice were housed under immunocompromised conditions with autoclaved cages, water, and irradiated feed. Mice were treated with either p40_2_ mAb or p40 mAb twice a week (2 mg/kg) via intraperitoneal injection. Control mice received only sterile saline. Mice were weighed one time per week, and observations were noted. Tumor size was measured three times per week as soon as the xenograft was palpable using a digital caliper and recorded in a lab notebook for each animal. To calculate tumor volume, we used the following formula:V = (length × width^2^)/2

Tumor volume was measured until experimental endpoints were reached, and once control tumors reached about 20 mm. National Institutes of Health guidelines were followed for animal experiments. All animal experiments were approved by the Institutional Animal Care and Use Committee (IACUC) of the Rush University Medical Center.

#### 2.9.1. Histopathology

IACUC protocol was followed to euthanize mice by carbon dioxide fixation followed by cervical dislocation. Mouse blood samples were taken via cardiac puncture and tumor tissue was excised post-euthanasia. Tumor tissue sections were fixed in 4% paraformaldehyde for at least 24 h, embedded in paraffin, and mounted on slides in 5 µm sections. Slides were stained with hematoxylin and eosin for additional morphological studies. The tumor area was analyzed as described by us in previous studies [21].

#### 2.9.2. Statistical Analysis

For tumor regression, a repeated measure ANOVA was conducted, and quantitative data were presented as the mean ± SEM. Statistical significance was assessed via one-way ANOVA with Tukey post hoc analysis. All further data were expressed as means ± SD of three independent experiments. A *p* value of less than 0.05 was considered statistically significant. Analyses were completed by GraphPad Prism 7.02 software.

## 3. Results

### 3.1. Levels of p40_2,_ p40 and IL-12, IL-23 in Serum of Pancreatic Cancer Patients

Earlier, it was not possible to examine the roles of p40_2_ and p40 in the pathogenesis of diseases due to the lack of specific neutralizing mAbs; therefore, we generated neutralizing mAbs against p40_2_ and p40 [17] and developed an ELISA to detect and quantify p40_2_ and p40 independently [17,18,19,20]. In previous studies, we found that levels of p40 are much higher in the serum of prostate cancer patients [18] and breast cancer patients [19] compared to healthy controls. In order to understand whether this observation is specific to prostate or breast cancer, we measured levels of p40_2_, p40, IL-12, and IL-23 in the serum of pancreatic cancer patients (n = 10) and healthy controls (n = 10). We used the serum of pre-treated pancreatic cancer patients, to eliminate any influence of different treatments on cytokine levels. Compared to our previous research in prostate or breast cancer cases, we observed that p40_2_ levels were substantially elevated in the serum of pancreatic cancer patients compared to healthy controls (Figure 1A). Similar to the observations in prostate or breast cancer, the level of p40 was also significantly higher in pancreatic cancer patients than in healthy controls (Figure 1B). Likewise, when comparing them to prostate and breast cancer cases, levels of IL-12 (Figure 1C) and IL-23 (Figure 1D) were also significantly lower in pancreatic cancer as compared to healthy controls.

To further understand the significance of our finding, we detected the levels of the p40 family of cytokines in human pancreatic cancer cell lines. Pancreatic cancer cells such as PANC-1 and BxPC-3 as well as normal human pancreatic duct epithelial (HPDE) [26] cells were cultured under serum-free conditions for 24 h, followed by measurements of p40_2_, p40, IL-12, and IL-23 by sandwich ELISA. Similarly to the serum of pancreatic cancer patients, levels of p40_2_ and p40 levels were higher in both PANC-1 and BxPC-3 pancreatic cancer cells as compared to normal HPDE cells (Figure 1E,F). In contrast, levels of IL-12 and IL-23 were remarkably lower in PANC-1 and BxPC-3 cells in comparison to HPDE cells (Figure 1G,H).

### 3.2. Immunotherapies with mAbs against p40_2_ and p40 Induces Death in Human Pancreatic Cancer Cells

Earlier in our studies, we examined the role of p40 in the survival of cancer cells, and found that in the p40 family of cytokines, only p40 monomer levels were observed to be higher than other subsets of this immune-modulating cytokine [18]. Contrary to prior observations, this study reveals the cytotoxic effects of p40_2_ mAb in human PDAC cell lines. As seen in (Figure 2A,B), p40_2_ mAb increased LDH [27] release in both PANC-1 and BxPC-3 cells, and a decrease in cell viability (Figure 2D,E), as shown by MTT metabolism, indicating the induction of cell death in PDAC cells by the neutralization of p40_2_ by p40_2_ mAb (a3-1d). This result is regarded as specific since control and hamster IgG had no effect on either LDH or MTT in human PDAC cells.

It is important to note that in HPDE cells (normal pancreatic cells), treatment with p40_2_ mAb had no significant effects on either LDH (Figure 2C) or MTT (Figure 2F). To confirm our previous observation, we performed FACS double-labeling with AnnexinV and propidium iodide (PI) and found that p40_2_ mAb treatment significantly increased apoptotic cells in both PANC-1 (Figure 2G) and BxPC-3 (Figure 2H) cells, followed by quantitative analyses (Figure 2J) and (Figure 2K). In comparison to the LDH and MTT results, treatment with p40_2_ mAb in HPDE cells displayed no significant effects (Figure 2I) as demonstrated by quantitative analyses (Figure 2L). As indicated in other cancer cell types by previous studies [18,19], the p40 mAb (a3-3a) increased LDH release (Figure 3A,B) and decreased MTT (Figure 3D,E) in PDAC cells. Compared with the p40_2_ mAb-treated cell results, p40 mAb also had no significant effects on normal pancreatic cells (Figure 3C,F).

Accordingly, FACS analysis with AnnexinV and PI in PANC-1 (Figure 3G) and BxPC-3 (Figure 3H) cells confirmed this observation and described that p40 mAb treatment also increased the level of apoptotic cells (Figure 3J,K) in PDAC cell lines. In line with previous results, treatment with p40 mAb in HPDE cells displayed no significant effects (Figure 3I,L). Moreover, TUNEL staining clearly showed that the population of TUNEL-positive cells in p40_2_ mAb-treated PDAC cells (Figure 4A,B) was larger (Figure 4G–H) than in the control, IgG, or normal pancreatic cells (Figure 4C,I). In p40 mAb-treated PDAC cells (Figure 4D,E), our TUNEL results clearly indicated increased TUNEL bodies (Figure 4J,K) compared to the control, IgG, or normal pancreatic cells (Figure 4F), where no presence (Figure 4L) of TUNEL bodies was visualized. Overall, evidence illustrates that neutralization of p40_2_ by p40_2_ mAb and functional blocking of p40 by p40 mAb stimulates cell death in PDAC cell lines, not normal pancreatic cells.

### 3.3. Upregulation of IFN-γ In Vitro in PDAC Cells after p40_2_ mAb and p40 mAb Treatments

The IL-12 signaling pathway induces IFN-γ production in various cell types once activated [12,18,19]. We investigated the mechanisms by which p40_2_ mAb and p40 mAb induced a death response in PDAC cancer cells. Earlier studies showed that higher expression of IFN-γ in p40 mAb-treated cancer cells was involved in cell death [28]. IFN-γ production is a confirmed therapeutic approach to stimulate cytotoxicity in cancer cells [29]. IFN-γ is a pleiotropic cytokine with antitumor and immunomodulatory functions; it plays a central role in orchestrating innate and adaptive immune responses [30]. IFN-γ production is controlled by IL-12-activated Th1 cells [31]. Although IFN-γ is a T-cell-derived cytokine, other cells such as epithelial cells [18] are capable of producing IFN-γ. Studies found that IFN-γ can induce apoptosis in tumor-specific T-cells, guiding antitumor immunity [32]. We examined if p40_2_ mAb and p40 mAb treatment caused the upregulation of IFN-γ in PDAC cells. Immunostaining analysis clearly indicated that p40_2_ mAb and p40 mAb treatments increased the level of IFN-γ in both PANC-1 (Figure 5A) and BxPC-3 (Figure 5B) cells, compared to either control cells or IgG-treated cells. Followed by counting of IFN-γ cells, quantitative analysis of both PANC-1 (Figure 5C) and BxPC-3 (Figure 5D) cells showed elevated levels of IFN-γ in p40_2_ mAb- and p40 mAb-treated PDAC cells.

To further confirm, the mRNA expression of IFNγ was checked by real-time PCR. We found increased IFN-γ mRNA expression in PANC-1 (Figure 5E) and BxPC-3 (Figure 5F) cells that were treated with p40_2_ mAb, but not control IgG. Likewise, p40 mAb, but not control IgG, upregulated IFN-γ mRNA expression in PANC-1 (Figure 5G) and BxPC-3 (Figure 5H) cells as compared to control untreated cells. To confirm it further, we performed dual FACS analysis with IFN-γ after live/dead cell exclusion and found that treatment with p40_2_ mAb and p40 mAb markedly increased apoptotic cells in both PANC-1 (Figure 5I) and BxPC-3 (Figure 5J) cells. This was corroborated by quantitative analysis (PANC-1, Figure 5K; BxPC-3, Figure 5L). Collectively, these results suggest that neutralization of p40_2_ by p40_2_ mAb and p40 by p40 mAb is capable of inducing cell death in PDAC cells.

### 3.4. Immunotherapies with p40_2_ mAb and p40 mAb Lead to Tumor Regression in a Patient-Derived Xenograft (PDX) Mouse Model of Pancreatic Cancer

The PDX model, reliably developing tumors within a few weeks, recapitulates the spatial structure of cancer, the intratumoral heterogeneity of cancer, and the genomic features of cancer patients [34,35]. We used the PDX model (ID# TM01212; Jackson Lab) [36] for this study. In this particular model, cryopreserved tissue obtained from patients with pancreatic cancer was minced and implanted subcutaneously into the NOD scid gamma (NSG) mouse’s fat pad. To examine the role of p40_2_ and p40 in the progression of PDAC tumors, we investigated the effect of p40_2_ mAb and p40 mAb on tumor size in PDX mice. Once palpable tumors were detected, they were measured three times weekly until experimental endpoints were reached.

Once tumors reached 0.5 cm^2^ in area, mice were treated with p40_2_ mAb (a3-1d) and p40 mAb (a3-3a) at a dose of 2 mg/kg body wt/week i.p. for 3.5 weeks (Figure 6A). When control or IgG tumors reached around 10% of mouse body weight, tumors were labeled with IR dye 800-conjugated 2-deoxyglucose via tail vein injection and imaged using a LI-COR Odyssey infrared scanner (Figure 6B). This was corroborated by quantitative analysis (Figure 6C). Notably, we observed that treatment with p40_2_ mAb and p40 mAb separately reduced tumor size as discerned from images of excised tumors (Figure 7A). Although we did not observe complete tumor regression of PDAC tumors with either p40_2_ mAb or p40 mAb, our tumor measurements, seen in the tumor regression curve, clearly showed that the p40_2_ mAb- and p40 mAb-treated mice had smaller tumors compared to either control or IgG-treated PDX mice (Figure 7B). H&E staining showed a substantial decrease in live cells and necrosis in both p40_2_ mAb- and p40 mAb-treated mice compared to either untreated control or IgG-treated mice (Figure 7C). These results were corroborated by quantification of the tumor area (Figure 7D).

### 3.5. Immunotherapies with p40_2_ mAb and p40 mAb Induce a Death Response and Upregulation of IFN-γ in Tumor Tissues of PDX Mouse Model of Pancreatic Cancer

Cancer cells have a notorious property of evading cell death [37]. Accordingly, we monitored programmed cell death or apoptosis in tumor tissues by TUNEL assay. For better visualization of apoptosis, tumor sections were double-labeled for TUNEL and cleaved caspase 3, an apoptotic marker capable of propagating an apoptotic signal through enzymatic activity [38]. As expected, very few TUNEL-positive (Figure 8A,B) or cleaved caspase 3-positive (Figure 8A,C) cells were found in tumors of untreated PDX mice. However, we observed a significant increase in TUNEL-positive (Figure 8A,B) and cleaved caspase 3-positive (Figure 8A,C) cells in tumors of p40_2_ mAb- and p40 mAb-treated PDX mice as compared to untreated PDX mice (Figure 8A). However, IgG treatment remained unable to stimulate either TUNEL-positive (Figure 8A,B) or cleaved caspase 3-positive (Figure 8A,C) cells in tumors of PDX mice (Figure 8A), indicating the specificity of treatment.

Since upregulation of IFN-γ production is one of the key mechanisms in tumor regression, we monitored the levels of IFN-γ levels in PDAC tumors by double-labeling tumor tissues with IFNγ and CA19-9 (Figure 8D), a commonly used PDAC diagnostic marker used to detect pancreatic cancer treatment [39]. As expected, tumor tissues of untreated PDX mice expressed CA19-9 (Figure 8D,F), but not IFNγ (Figure 8D,E). On the other hand, treatment of PDX mice with both p40_2_ mAb and p40 mAb led to a significant decrease in CA19-9 (Figure 8D,F) and a marked increase in IFNγ (Figure 8D,E) in tumor tissues, indicating an improvement in disease prognosis in comparison to either untreated control or IgG-treated PDX mice.

Several molecules such as Bcl-2 and a phosphorylated version of BAD are known to support cell survival and anti-apoptosis [40,41]. Therefore, to further confirm our findings on cell survival and death in tumor tissues of PDX mice, we performed immunoblot analyses of phospho-BAD and Bcl-2 (Figure 8G and Appendix A) in tumors of in p40_2_ mAb- and p40 mAb-treated PDX mice. For raw blots, please see Figure 1. Consistent with the induction of apoptosis, we found a significant decrease in phospho-BAD (Figure 8G,H) and Bcl-2 (Figure 8G,I) in the tumor tissues of p40_2_ mAb- and p40 mAb-treated PDX mice as compared to those of untreated PDX mice. Again, these results were specific, as IgG treatment remained unable to decrease the levels of phospho-BAD (Figure 8G,H) and Bcl-2 (Figure 8G,I) in the tumor tissues of PDX mice. Overall, these results suggest that, similarly to the observations in PDAC cells, p40_2_ mAb and p40 mAb treatments are capable of inducing apoptosis in vivo in PDAC tumors in PDX mice.

## 4. Discussion

PDAC is the most common pancreatic neoplasm, accounting for more than 90% of all pancreatic cancers. The overall survival rate is 12%, and it remains the third leading cause of cancer deaths [42]. Treatment is largely determined by the stage of disease, and unfortunately, most cases present with locally advanced or non-resectable stages and metastases [43]. The current standard of care is surgical resection followed by adjuvant chemotherapy and/or radiation [1,3]. Monoclonal antibodies (mAbs) have proven effective at targeting solid tumors for decades [44]. Accordingly, although multiple immunotherapies such as anti-CTLA-4, or anti-PD-1 [45] are being used for PDAC treatment, there is an urgent need for the development of effective therapeutic strategies for PDAC.

The p40 family of cytokines, representing the first line of the immune response, plays a key role in coordinating the activities of innate and adaptive immune systems [8,18,20,22]. Recently, we have described an increase in p40 and decreases in p40_2_, IL-12, and IL-23 in the serum of patients with prostate and breast cancers [18,19]. Accordingly, the selective neutralization of p40 by p40 mAb leads to the regression of prostate cancer and triple-negative breast cancer in mouse models [18,19]. Similarly to prostate and breast cancer cases, the present study highlights the upregulation of p40_2_ in the serum of pancreatic cancer patients as compared to healthy controls and in supernatants of human PDAC cells in comparison to normal human HPDE pancreatic cells. This is the first demonstration of p40_2_ upregulation in any cancer. In addition to p40_2_ and similarly to prostate and breast cancer, we have also seen an increase in p40 in the serum of pancreatic cancer patients and human PDAC cells as compared to their respective controls, indicating the possible involvement of p40_2_ and p40 in the pathogenesis of pancreatic cancer. *First*, the levels of p40_2_ and p40 were significantly higher in human PDAC cells compared with IL-12, IL-23, and notably, normal pancreatic cells. *Second*, the neutralization of p40_2_ and p40 induced cell death in PDAC cells, but more importantly, it did not in normal pancreatic cells. *Third*, our Annexin-V staining experiments demonstrated higher levels of death in PDAC cells compared to normal pancreatic cells. *Fourth*, our TUNEL staining also indicated the same findings. *Lastly*, intraperitoneal injection of p40_2_ mAb and p40 mAb, but not IgG, significantly reduced the sizes of tumors grown in PDAC PDX mice. Overall, our study delineates an important role of p40_2_ and p40 in the progression/regression of pancreatic cancer.

For accomplishing tumor regression, it is obligatory to stimulate apoptosis in tumor tissues. We demonstrated, from multiple approaches, the substantial death response in PDAC tumors after treatment with p40_2_ mAb and p40 mAb. Our conclusions are based on the following observations: Initially, tumor growth in mice was measured, and visualized using IR dye. Then, H&E staining showed decreased tumor size, and a hollow core in p40_2_ mAb- and p40 mAb-treated mice compared to untreated or control IgG-treated PDX mice. Next, the number of TUNEL-positive cells was much higher in tumors treated with p40_2_ mAb and p40 mAb compared to either untreated control or IgG-treated PDX mice. Finally, in p40_2_ mAb- and p40 mAb-treated tumors, we observed a decrease in anti-apoptotic markers, phospho-BAD, and Bcl-2, a common survival protein. Therefore, both p40_2_ mAb and p40 mAb may be a considerable therapy for stimulating a cell death response in PDAC tumors.

While investigating mechanisms responsible for p40_2_- and p40-mediated tumor cell death, we examined upregulation of IFN-γ, an important cytotoxic cytokine [46], when treated with p40_2_ mAb. IFN-γ production is commonly motivated by IL-12, and elevated levels are indicative of tumor regression [12,47,48]. As observed in prostate [18] and breast cancer cells [12] with the p40 mAb, IFN-γ production was elevated when PDAC cells were treated with both p40_2_ mAb and p40 mAb. It is likely that p40_2_ mAb and p40 mAb mediated death in PDAC cells, and tumor tissue was stimulated by the cytotoxic quality of IFN-γ. The suppressed tumor growth in vivo validated that p40_2_ mAb and p40 mAb induce PDAC cell death via IFN-γ. To monitor the effectiveness of p40_2_ mAb and p40 mAb from another perspective, we observed the levels of carbohydrate antigen (CA19-9), a commonly used tumor marker for PDAC. This glycoprotein complex, produced by ductal cells in the pancreas, is expressed in both benign and malignant disorders [49]. It is currently used to monitor treatment response for resectability and prognosis [50]. We observed exceedingly decreased levels of CA19-9 in p40_2_ mAb and p40 mAb tumor sections. Consistent with the cytotoxic nature of IFN-γ, our results further demonstrate that treatment with p40_2_ mAb and p40 mAb induces cell death and apoptosis in PDAC.

## 5. Limitations

This therapeutic approach involves mAb treatment, and the efficacy may vary from patient to patient depending on the level of p40_2_ and/or p40. At present, we do not know whether the neutralization of p40_2_ and p40 will cause immune suppression or not. Like any other medications, there could be side effects, which are not known at present. Therefore, further studies are necessary to ascertain the safety profile over time. Moreover, there is a possibility that this therapy may interact with other medications. In that case, further data are necessary to determine additional contraindications.

## 6. Conclusions

To summarize, our studies demonstrated higher levels of p40_2_ and p40 in the serum of pancreatic cancer patients and supernatants of human PDAC cells in comparison to their respective controls. Accordingly, functional blocking of p40_2_ and p40 by p40_2_ mAb and p40 mAb, respectively, caused apoptosis and cell death in human PDAC cells, but not in healthy pancreatic cells. In the PDX mouse model of pancreatic cancer, treatments with p40_2_ mAb and p40 mAb, separately, also resulted in apoptosis and tumor regression. Although the PDX mouse model of pancreatic cancer may not perfectly reproduce the in vivo condition of a pancreatic adenocarcinoma tumor in human pancreatic cancer patients, our results suggest that the p40_2_ mAb and p40 mAb may provide new immunotherapeutic options against pancreatic cancer.

## Figures and Tables

**Figure 1 cancers-15-05796-f001:**
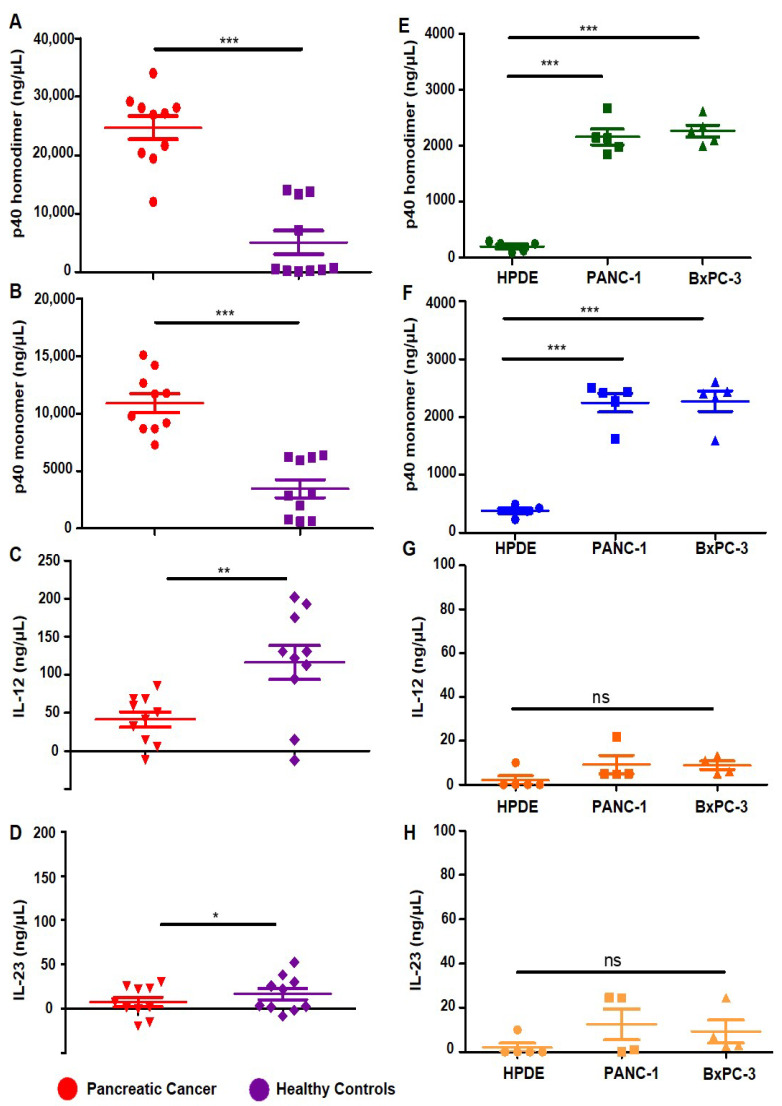
Levels of p40_2_, p40, IL-12, and IL-23 in serum of pancreatic cancer patients and pancreatic cancer cells. We checked levels of the p40 (IL-12) family of cytokines in serum of pre-treated pancreatic cancer patients (n = 10) (red) and age-matched healthy controls (n = 10) (purple). The samples were purchased from Discovery Life Sciences (Los Osos, CA, USA) and were measured for p40_2_ (**A**), p40 (**B**), IL-12 (**C**), and IL-23 (**D**) by sandwich ELISA. Supernatants of HPDE normal pancreatic cells compared to PANC-1 and BxPC-3 pancreatic cancer cells (**E**,**F**) were analyzed for levels of p40_2_, and p40 by sandwich ELISA along with levels of IL-12 and IL-23 (**G**,**H**). Results are mean + SD of three different experiments. * *p* < 0.05; ** *p* < 0.01; *** *p* < 0.001.

**Figure 2 cancers-15-05796-f002:**
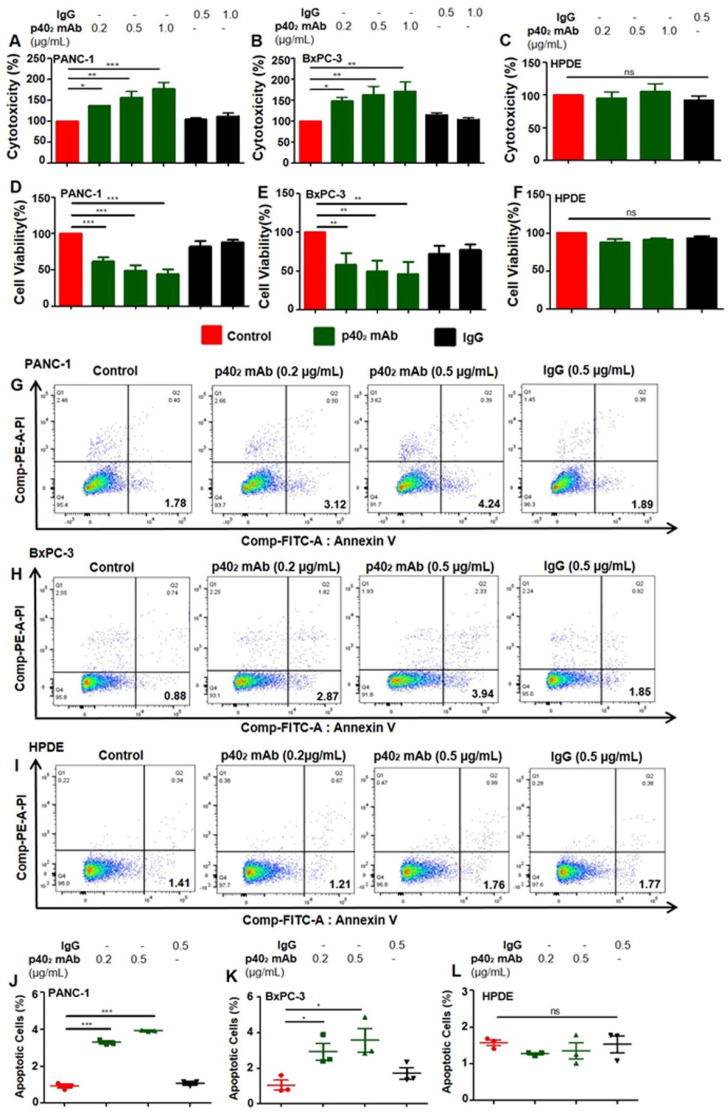
Monoclonal antibody-mediated neutralization of p40_2_ by p40_2_ mAb induces apoptosis in PANC-1 and BxPC-3 pancreatic cancer cells but not in normal pancreatic cells. Cells were treated for 24 h under serum-free condition followed by detecting LDH release (**A**–**C**) and MTT metabolism (**D**–**F**). Single-cell suspensions of pancreatic cancer cells and normal cells (**G**–**I**) were studied by dual FACS for Annexin V-FITC and propidium iodide (PI) using the FACS-Canto II system (BD Biosciences) followed by analysis using FlowJo Software (v10). Quantification of the percent of apoptotic cells (PI^−^ Annexin V^+^ early apoptotic) (**J**–**L**). Results are mean + SD of three different experiments. * *p* < 0.05; ** *p* < 0.01; *** *p* < 0.001.

**Figure 3 cancers-15-05796-f003:**
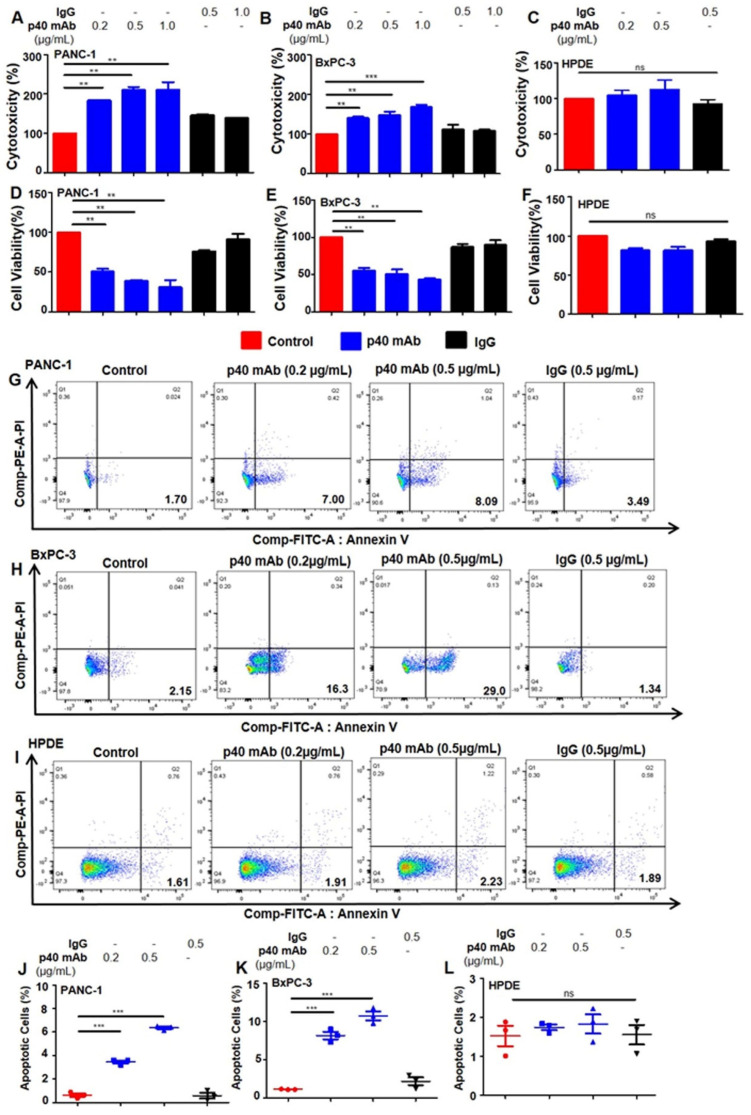
Monoclonal antibody-mediated neutralization of p40 by p40 mAb induces apoptosis in PANC-1 and BxPC-3 pancreatic cancer cells but not in normal pancreatic cells. Cells were treated for 24 h under serum-free condition followed by detecting LDH release (**A**–**C**) and MTT metabolism (**D**–**F**). Single-cell suspensions of pancreatic cancer cells and normal cells (**G**–**I**) were studied by dual FACS for Annexin V-FITC and propidium iodide (PI) using the FACS-Canto II system (BD Biosciences) followed by analysis using FlowJo Software (v10). Quantification of the percent of apoptotic cells (PI^−^ Annexin V^+^ early apoptotic) (**J**–**L**). Results are mean + SD of three different experiments. ** *p* < 0.01; *** *p* < 0.001; ns, not significant.

**Figure 4 cancers-15-05796-f004:**
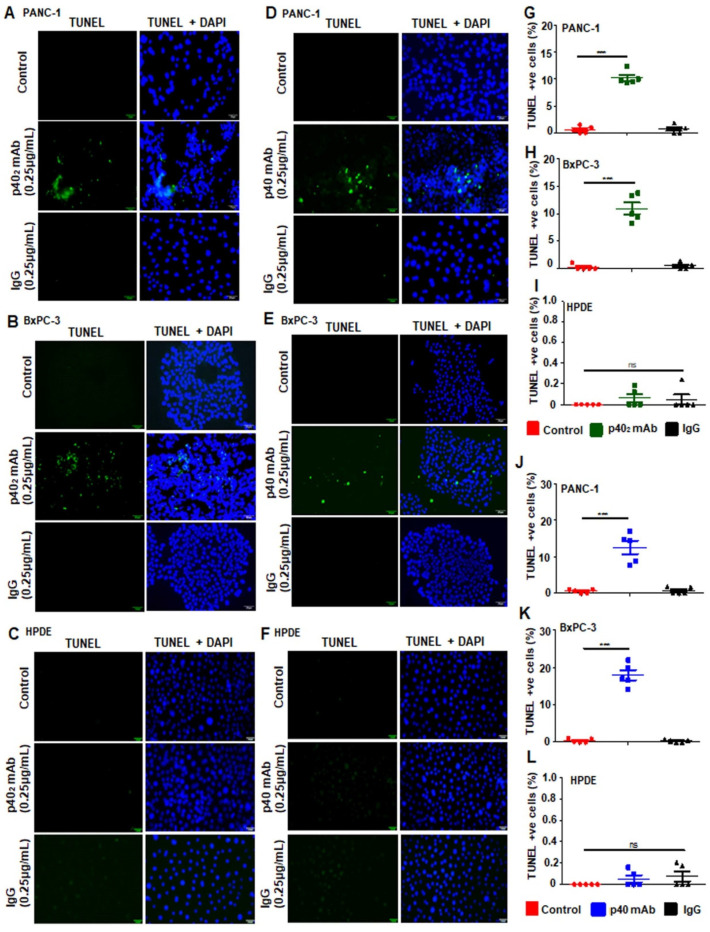
Increase in death response by p40 homodimer- and p40 monomer-treated pancreatic cancer cells but not in normal pancreatic cells. Cells were labeled for TUNEL in control cells and pancreatic cancer cells treated with IgG, p40 homodimer (**A**,**B**), and p40 monomer (**D**,**E**), and normal pancreatic cells (**C**,**F**). TUNEL-positive cells were counted and quantified in a 20× field in p40 homodimer-treated (**G**,**H**), in p40 monomer (**J**,**K**) -treated PDAC cells, and in normal pancreatic cells (**I**,**L**) as percent of control. TUNEL-positive cells were counted in 10 different images per group and plotted as a percentage of control. Results are mean + SD of three different experiments. *** *p* < 0.001; ns, not significant. Scale bar = 20 µm.

**Figure 5 cancers-15-05796-f005:**
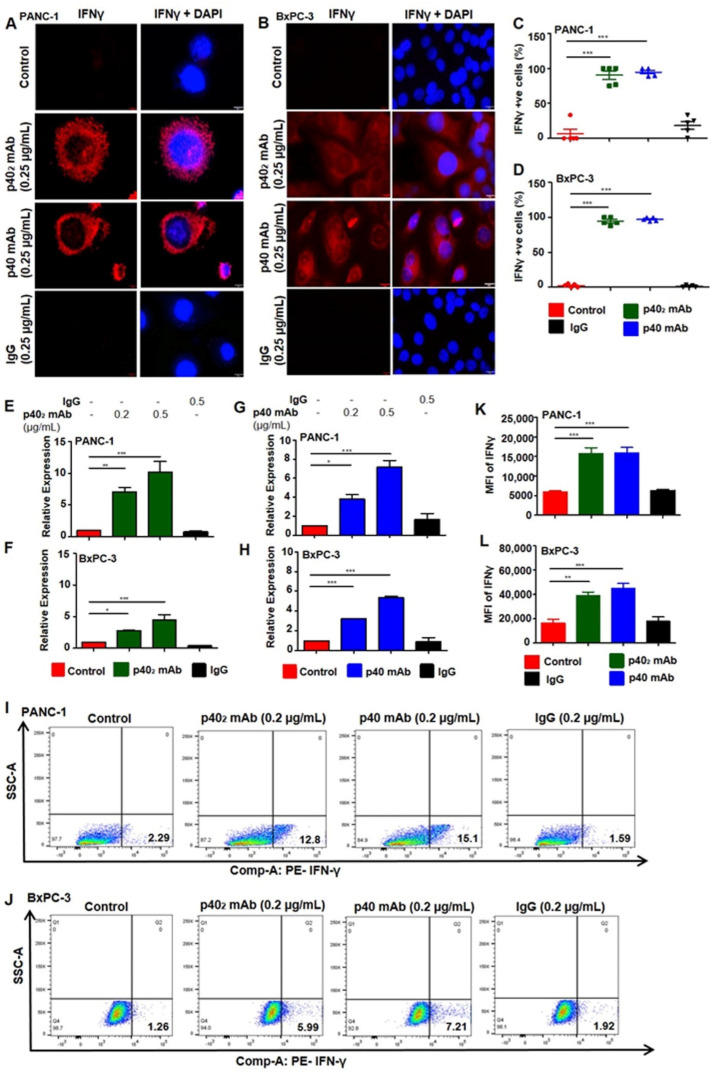
Neutralization of p40_2_ and p40 increases levels of IFN-γ in pancreatic cancer cells. Immunohistochemical analysis at 60× magnification of PANC-1 (**A**) and BxPC-3 cells (**B**) treated with p40 homodimer or p40 monomer were labeled with IFN-γ (red). DAPI was used to stain the nuclei. Cells positive for IFN-γ were counted and quantified (5 images per slide) [33] (**C**,**D**). Real-time mRNA analysis of IFN-γ levels in pancreatic cancer cells treated with p40 homodimer (**E**,**F**) and p40 monomer (**G**,**H**). Single-cell suspensions isolated from PANC-1 (**I**) and BxPC-3 (**J**) cells were studied by dual FACS for IFN-γ and live/dead using the FACS Canto II system (BD Biosciences) followed by analysis using FlowJo Software (v10.8). PDAC cell quantification of early apoptotic cells (PI-Annexin V+) treated with p40_2_ mAb and p40 mAb (**K**,**L**). Results are mean + SD of three different experiments. * *p* < 0.05; ** *p* < 0.01; *** *p* < 0.001. Scale bar = 10 µm.

**Figure 6 cancers-15-05796-f006:**
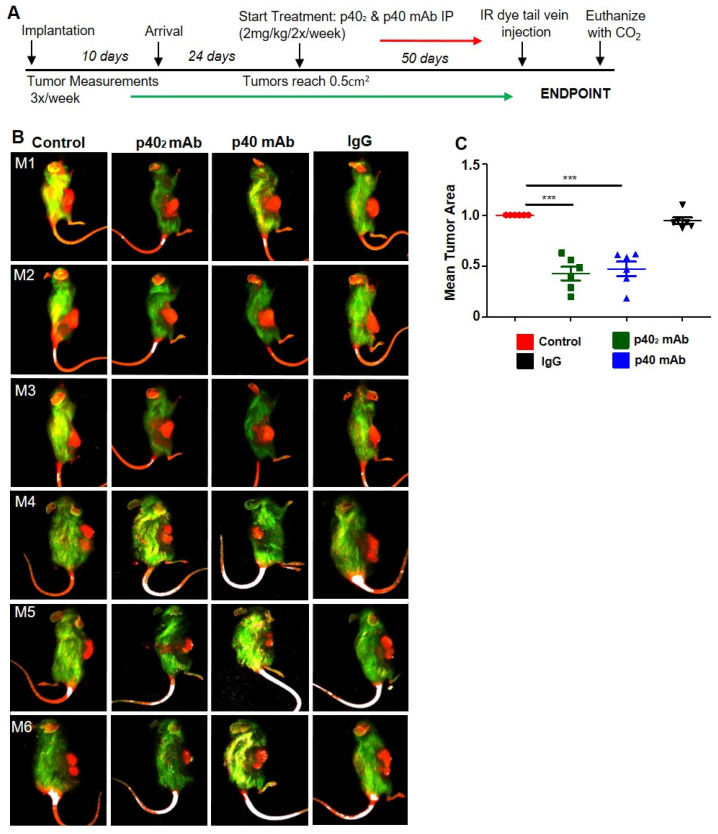
PDAC tumor regression in vivo in PDX mice treated with mAbs against p40 homodimer and p40 monomer. The experimental design is illustrated for PDAC PDX mice (**A**). Female 6–8-week-old NOD scid gamma (NSG) mice were engrafted with PDAC tumor fragments (TM01212) in the flank. When PDX mice (n = 6) tumors reached 0.5 cm^2^, mice were treated with p40_2_ and p40 mAbs and control hamster IgG at a dose of 2 mg/kg/body weight twice a week. After four weeks of treatment, tumors were labeled with Alexa800-2DG dye via tail vein injection and imaged using LICOR Odyssey infrared imaging system (**B**). This was corroborated by quantitative analysis (**C**). Six mice (n = 6) were included in each group. Results are mean + SD of three different experiments. *** *p* < 0.001.

**Figure 7 cancers-15-05796-f007:**
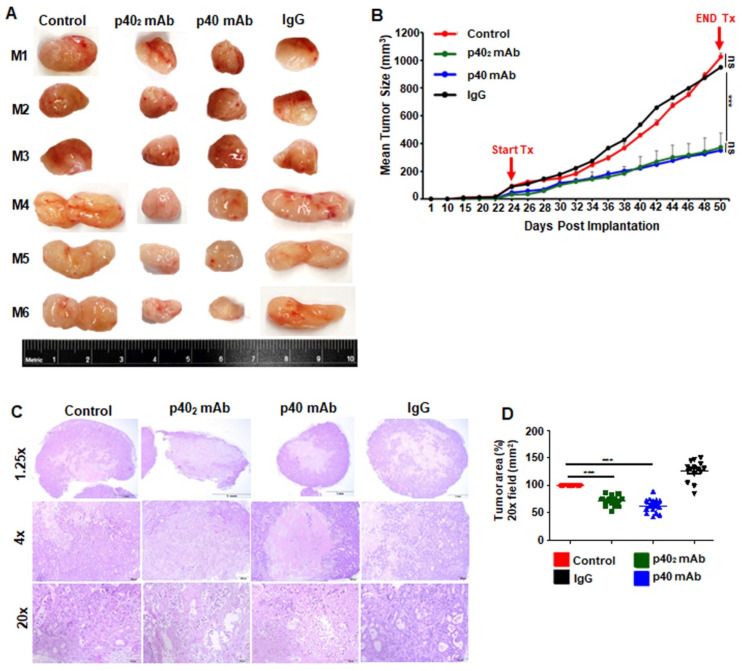
Results were compared with control and IgG groups. Tumors were surgically removed from the flanks of all mice. Six mice (n = 6) were included in each group (**A**). Tumor size was monitored every other day (**B**). Tumor sections were stained with H&E (**C**). The histological tumor area was quantified in a 20× field as percent of control (**D**). Results are mean + SD of three different experiments. *** *p* < 0.001; ns, not significant. Scale bar = 1 mm, 200 µm, and 50 µm.

**Figure 8 cancers-15-05796-f008:**
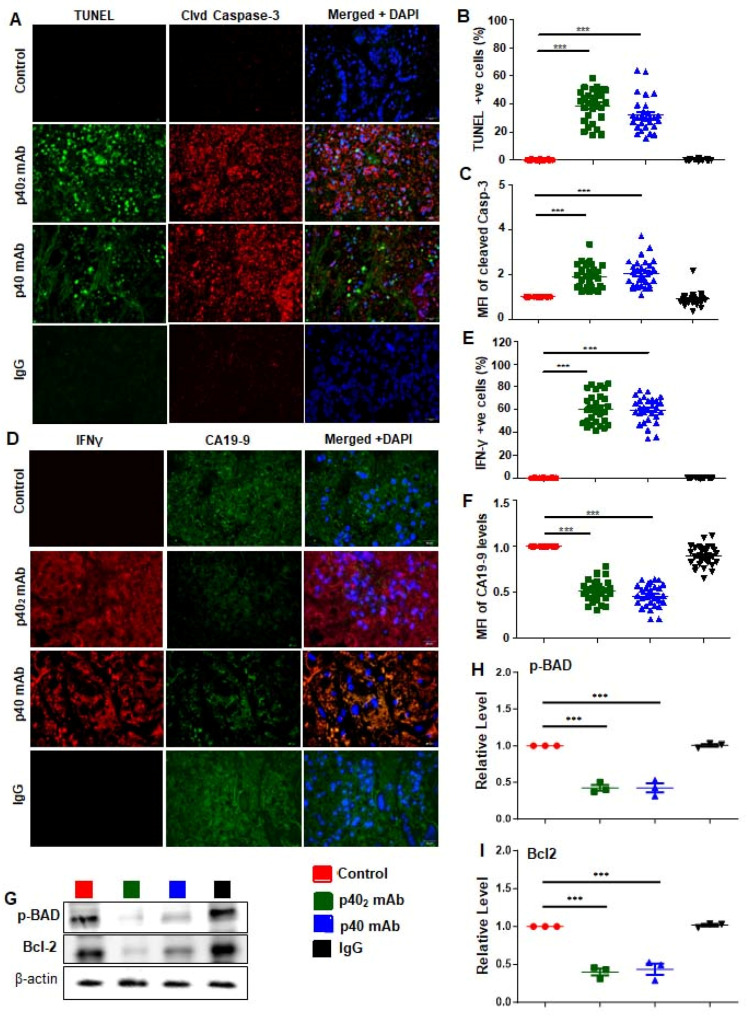
Stimulation of a death response and IFN-γ induction in PDAC tumors of PDX mice by p40_2_ and p40 mAbs. After 4 weeks of treatment, tumor sections were double-labeled for TUNEL and cleaved caspase-3 (**A**) followed by counting TUNEL-positive cells in 5 sections of 6 mice per group (**B**), and mean fluorescent intensity (MFI) of cleaved caspase 3 (**C**) was quantified in a 40× field as percent of control. Tumor cross sections were double-immunolabeled for IFN-γ and CA19-9, a prognostic PDAC marker (**D**). Cells positive for IFN-γ (**E**) and with a MFI of CA19-9 (**F**) were quantified in a 60× field as percent of control was counted (5 images per mouse) of six different mice per group. DAPI was used to stain the nuclei. The stained tumor sections were quantified in a 40× field as percent of control. Tumor tissues were immunoblotted for survival proteins (phospho-BAD and Bcl-2) (**G**). Actin was run as a loading control. Results represent three independent experiments. Bands were scanned and values are presented as relative to control; the relative density of immunoblot analyses was normalized with β-actin (**H**,**I**). Results are mean + SD of three different experiments. *** *p* < 0.001. Scale bar = 40 µm.

**Table 1 cancers-15-05796-t001:** Cell lines for this project.

Cell Line	Diagnosis	Mutation	Cell Type	Tissue	Age	Sex	Manufacturer
HPDE-H6c7	Normal	wt p53 wt KRAS	Epithelial	Pancreatic duct	63	F	Kerafast
PANC-1	Epithelioid Carcinoma	mut p53 mut KRAS	Epithelial	Pancreatic duct	56	M	ATCC
BxPC-3	Adenocarcinoma	mut p53 wt KRAS	Epithelial	Pancreas	61	F	ATCC

**Table 2 cancers-15-05796-t002:** Serum Samples used in this study.

Patient ID	Diagnosis	Age	Sex	Ethnicity	Manufacturer
**Healthy Controls**					
HC-130	Normal	39	F	White	DLS
HC-798	Normal	57	F	White	DLS
HC-144	Normal	65	F	White	DLS
HC-792	Normal	51	F	White	DLS
HC-594	Normal	37	F	American Indian	DLS
HC-806	Normal	34	F	White	RUMC
HC-593	Normal	63	M	White	RUMC
HC-810	Normal	63	M	White	RUMC
HC-900	Normal	54	M	Asian	RUMC
HC-901	Normal	32	F	Asian	RUMC
**Pancreatic Cancer**					
PC-416	Pancreatic Cancer	65	F	Black	DLS
PC-316	Adenocarcinoma	58	F	White	DLS
PC-716	Adenocarcinoma	68	M	Black	DLS
PC-617	Adenocarcinoma	78	M	White	DLS
PC-717	Pancreatic Cancer	71	M	White	DLS
PC-2617	Adenocarcinoma	52	M	White	DLS
PC-217	Adenocarcinoma	58	M	White	DLS
PC-417	Adenocarcinoma	57	F	Black	DLS
PC-110	Adenocarcinoma	74	F	White	DLS
PC-418	Adenocarcinoma	70	M	Black	DLS

**Table 3 cancers-15-05796-t003:** Antibodies used in this study.

Antibody	Manufacturer	Catalog	Host	Application	Dilution
Cleaved caspase 3	Cell Signaling	#9661	rabbit	IF	1:1000
IFN-γ	Invitrogen	MA5-44024	mouse	IF	1:200
CA19-9	Invitrogen	MA5-12421	mouse	IF	1:200
Phospho-BAD	Santa-Cruz	sc-166932	mouse	WB	1:200
Bcl-2	Santa-Cruz	sc-7382	mouse	WB	1:200

IF, Immunofluorescence; WB, Western blotting.

## Data Availability

All data are present in this manuscript.

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
