# Peer review of "Neutralization of p40 Homodimer and p40 Monomer Leads to Tumor Regression in Patient-Derived Xenograft Mice with Pancreatic Cancer"

_cancers, 2023, doi:10.3390/cancers15245796_

Round 1

Reviewer 1 Report

Comments and Suggestions for Authors

The authors examine the expression of p402 and p40 in pancreatic cancer patient serum. They examine the application of the antibodies to the two molecules for therapeutic application in cells and in vivo.

The manuscript would be improved greatly if the authors would add scale bars to the immunofluorescent images.

Examining the existing cancer databases on the expression of the molecules in cancer repositories is important and will help in future personalization attempts.

If data on the two molecules is available in the cancer repositories, then stratifying the disease outcome by driver mutations may give insight into the target patients who may benefit from this treatment.

Comments on the Quality of English Language

Minor revisions.

Author Response

Reviewer 1:

Comment: The authors examine the expression of p402 and p40 in pancreatic cancer patient serum. They examine the application of the antibodies to the two molecules for therapeutic application in cells and in vivo.

The manuscript would be improved greatly if the authors would add scale bars to the immunofluorescent images.

Response: Scale bars are on all the merged images. Thanks.

Comment: Examining the existing cancer databases on the expression of the molecules in cancer repositories is important and will help in future personalization attempts.

If data on the two molecules is available in the cancer repositories, then stratifying the disease outcome by driver mutations may give insight into the target patients who may benefit from this treatment.

Response: There is no information on p40 monomer and p40 homodimer in existing cancer databases. Therefore, serum from cancer patients could be checked for the levels of p40 monomer and p40 homodimer before initiating personalized immunotherapy with p40 mAb and/or p40-2 mAb. Thanks.

All Changes are highlighted in yellow.

Reviewer 2 Report

Comments and Suggestions for Authors

Overall, this is a well written and executed manuscript which contributes insights into the potential of utilizing cytokines as an alternative modality for immunotherapy in the management of pancreatic cancer. Shortcomings of the manuscript have been well laid by authors, which could be used as potential long term future studies. Although the authors have successfully investigated and demonstrated the neutralization of p402 and p40 with use of mAbs there are some minor changes that have been suggested and could benefit for the overall publication quality.

Recommendations for authors:

1.       I am unsure if there is any need for Simple summary and an Abstract. My recommendation would be to stick to ONLY one, preferably the Abstract.

2.       I would recommend having the manuscript be reviewed for proper English editing. Few observations as for Line 58, I believe the authors suggest similar observations of elevated p402 and p40 were made in breast, prostate cancer as observed in pancreatic cancer patients, in comparison to their controls. The authors have used “In contrast” which means the complete reverse of what is suggested.

3.       Same thing as the above comment. Line 232 “in contrast”, Line 234, 455 “On the other hand”, Line 236 “However”, they all suggest pancreatic cancer versus breast and prostate cancer observations were different, which is not true. Please correct.

4.       Just curious, why were the cells cultured in serum free media?

5.       Were any biophysical properties analyzed to estimate the binding ratio.

6.       Hypothetically, if the authors had access to all analytical tools what biophysical properties could be analyzed to ensure the quality, reproducibility and/or stability of these complexes are maintained from batch-to-batch production.

7.       What are some challenges that could be anticipated if the therapy is approved for clinical trials.

Author Response

Reviewer 2:

Comment: Overall, this is a well written and executed manuscript which contributes insights into the potential of utilizing cytokines as an alternative modality for immunotherapy in the management of pancreatic cancer. Shortcomings of the manuscript have been well laid by authors, which could be used as potential long term future studies. Although the authors have successfully investigated and demonstrated the neutralization of p402 and p40 with use of mAbs there are some minor changes that have been suggested and could benefit for the overall publication quality.

Response: Thanks for nice and constructive comments.

Recommendations for authors:

  1. I am unsure if there is any need for Simple summary and an Abstract. My recommendation would be to stick to ONLY one, preferably the Abstract.

Response: According to “Instructions to Authors” and published articles, ‘Cancers’ requires both Simple summary and Abstract. Thanks.

  1. I would recommend having the manuscript be reviewed for proper English editing. Few observations as for Line 58, I believe the authors suggest similar observations of elevated p402 and p40 were made in breast, prostate cancer as observed in pancreatic cancer patients, in comparison to their controls. The authors have used “In contrast” which means the complete reverse of what is suggested.

Response: We have fixed this and the manuscript has been thoroughly edited. Thanks.

  1. Same thing as the above comment. Line 232 “in contrast”, Line 234, 455 “On the other hand”, Line 236 “However”, they all suggest pancreatic cancer versus breast and prostate cancer observations were different, which is not true. Please correct.

Response: We have corrected this. Thanks.

  1. Just curious, why were the cells cultured in serum-free media?

Response: Serum contains different hormones and growth factors that are capable of influencing cellular signaling pathways. Therefore, during treatment, cells including the control wells were kept under serum-free conditions. Thanks.

  1. Were any biophysical properties analyzed to estimate the binding ratio.

Response: Thanks for the suggestion. However, any biophysical properties of p40 monomer and p40 homodimer were not analyzed to estimate the binding ratio.

  1. Hypothetically, if the authors had access to all analytical tools what biophysical properties could be analyzed to ensure the quality, reproducibility and/or stability of these complexes are maintained from batch-to-batch production.

Response: Whenever we prepare fresh batch of mAb from a particular hybridoma that was thoroughly characterized earlier by single cell cloning, binding and functional blocking assay, we check its specificity by direct ELISA. Thanks.

Whenever needed, for other studies, we have performed many biophysical assays such as surface plasmon resonance (SPR), thermal shift assay (TSA), time-resolved fluorescence resonance energy transfer (TR-FRET), etc.  

  1. What are some challenges that could be anticipated if the therapy is approved for clinical trials.

Response: Some challenges that could be anticipated include but are not limited to finding eligible participants meeting the criteria of the trial, especially for an invasive malignancy such as pancreatic cancer. Any patient response variability as well as autoimmune adverse effects from the therapy would have to be considered as well. Tumor resistance and ability to evade immunotherapy is also problematic. The durability of the response and potential long-term effects of the immunotherapy must be taken into consideration. The cost and manufacturing challenges must also be taken into consideration.

All Changes are highlighted in yellow.

Reviewer 3 Report

Comments and Suggestions for Authors

1. The introduction offers a solid background on pancreatic cancer and its challenges. It might be beneficial to include a brief mention of the current treatment landscape, the limitations of existing therapies, and the need for innovative approaches like immunotherapy.

2. While you mention the use of monoclonal antibodies (mAbs) in the study, consider briefly explaining what mAbs are and how they work to target specific molecules in the immune system.

3. Having data related to the pancreatic tumor microenvironment would be helpful in the context of the research paper.

4. Quantitative data is required for Figure 6B.

5. Scale bar is required for Figure 6B.

Comments on the Quality of English Language

The English language in the provided text is generally clear and understandable, but there are a few areas where you can improve clarity and readability. Here are some suggestions: 

Check for consistency in terms of using "mAb" (monoclonal antibody). You can spell it out as "monoclonal antibody" the first time and then use "mAb" in parentheses, which is a common practice.

Author Response

Reviewer 3:

  1. The introduction offers a solid background on pancreatic cancer and its challenges. It might be beneficial to include a brief mention of the current treatment landscape, the limitations of existing therapies, and the need for innovative approaches like immunotherapy.

Response: We have done that. Thanks.

  1. While you mention the use of monoclonal antibodies (mAbs) in the study, consider briefly explaining what mAbs are and how they work to target specific molecules in the immune system.

Response: We have mentioned that in the text. Thanks.

  1. Having data related to the pancreatic tumor microenvironment would be helpful in the context of the research paper.

Response: We have provided data on IFNγ, CA19-9, cleaved caspase 3 from the pancreatic tumor microenvironment. Please see Figure 8A-I. Thanks.

  1. Quantitative data is required for Figure 6B.

Response: We have done that. Please see Figure 6C. Thanks.

  1. Scale bar is required for Figure 6B.

Response: It is difficult to add scale bar to this figure as it is from whole mice and showing in mm. Probably, due to this reason, any of the published manuscripts on infra-red imaging of mouse tumors did not show scale bars on whole mouse infra-red imaging pictures. However, we have shown scale bars for isolated tumors. Please see Figure 7A. Thanks.

All Changes are highlighted in yellow.

Round 2

Reviewer 3 Report

Comments and Suggestions for Authors

I appreciate your efforts in revising the manuscript. Your acceptance after the minor revision is noted. I will promptly proceed with addressing the specified corrections to minor methodological errors and text editing. If you have any additional comments or specific areas of focus, please feel free to let me know.

Thank you for your cooperation.

Comments on the Quality of English Language

The English language in the provided text is clear and comprehensible. The structure of sentences is generally well-organized, and the technical terms are appropriately used.